# Evaluation of 3D Human Intestinal Organoids as a Platform for EV-A71 Antiviral Drug Discovery

**DOI:** 10.3390/cells12081138

**Published:** 2023-04-12

**Authors:** Fatma Masmoudi, Nanci Santos-Ferreira, Dasja Pajkrt, Katja C. Wolthers, Jeroen DeGroot, Maria L. H. Vlaming, Joana Rocha-Pereira, Ludovico Buti

**Affiliations:** 1Charles River Laboratories, 2333 CR Leiden, The Netherlands; 2OrganoVIR Labs, Department of Medical Microbiology, Amsterdam UMC Location Academic Medical Center, Amsterdam Institute for Infection and Immunity, University of Amsterdam, 1105 AZ Amsterdam, The Netherlands; 3Laboratory of Virology and Chemotherapy, KU Leuven-Department of Microbiology, Immunology and Transplantation, Rega Institute, 3000 Leuven, Belgium; 4OrganoVIR Labs, Pediatric Infectious Diseases, Emma Children’s Hospital, Amsterdam UMC Location University of Amsterdam, 1105 AZ Amsterdam, The Netherlands

**Keywords:** enterovirus, organoids, antivirals, intestinal

## Abstract

Enteroviruses are a leading cause of upper respiratory tract, gastrointestinal, and neurological infections. Management of enterovirus-related diseases has been hindered by the lack of specific antiviral treatment. The pre-clinical and clinical development of such antivirals has been challenging, calling for novel model systems and strategies to identify suitable pre-clinical candidates. Organoids represent a new and outstanding opportunity to test antiviral agents in a more physiologically relevant system. However, dedicated studies addressing the validation and direct comparison of organoids versus commonly used cell lines are lacking. Here, we described the use of human small intestinal organoids (HIOs) as a model to study antiviral treatment against human enterovirus 71 (EV-A71) infection and compared this model to EV-A71-infected RD cells. We used reference antiviral compounds such as enviroxime, rupintrivir, and 2′-*C*-methylcytidine (2′CMC) to assess their effects on cell viability, virus-induced cytopathic effect, and viral RNA yield in EV-A71-infected HIOs and cell line. The results indicated a difference in the activity of the tested compounds between the two models, with HIOs being more sensitive to infection and drug treatment. In conclusion, the outcome reveals the value added by using the organoid model in virus and antiviral studies.

## 1. Introduction

Enterovirus A71 (EV-A71) is a small non-enveloped virus with a positive-sense single-stranded RNA genome that belongs to the genus *Enterovirus* of the *Picornaviridae* family [1]. EV-A71 is the main cause of hand, foot, and mouth disease (HFMD) in children. In addition, infection with EV-A71 can cause a range of symptoms, from mild respiratory and gastrointestinal symptoms to severe symptoms, such as neurological and cardiopulmonary complications that can lead to death [2]. EV-A71 is highly infectious and contagious, and since its discovery in 1969, many outbreaks have been reported around the world [3,4]. Like other enteroviruses, EV-A71 is transmitted via respiratory droplets, by the fecal-oral route, and through direct contact with virus-contaminated surfaces [5]. The gastrointestinal tract is considered the main site of EV-A71 replication [6]. Historically, immortalized cell lines, such as rhabdomyosarcoma (RD) or colonic adenocarcinoma (Caco-2) cells, have been used to model the EV-A71 infection in vitro [7,8]. These models have allowed the identification of several receptors for EV-A71 entry and replication [9,10,11]. In addition, immortalized cell lines have been used as a primary platform to test antiviral drugs against EV-A71 infections. Although these studies have greatly facilitated our understanding of the virus replication cycle in vitro, these models do not fully recapitulate the complexity of the human gastrointestinal epithelium with its multiple cell types [12]. Because cell lines constitute a single cell type, these models are limited and often fail to provide a reliable understanding of viral pathogenesis in humans [13]. In addition, the lack of a reliable in vitro, or ex vivo, model has hindered the research on novel antiviral treatment. Of note, there are no approved therapeutics to date for the treatment or prevention of EV-A71 infection.

The revolution in understanding cell development and differentiation has led to the generation of human three-dimensional (3D) organoid models. Organoids are generally defined by the presence of multiple-tissue-specific cell types derived from stem cells that self-organize in 3D [14]. For instance, the intestinal epithelium is populated in vivo by adult stem cells (Lgr5+) from which absorptive (enterocytes) and secretory (Paneth, enteroendocrine, and goblet) cell lineages are derived [15]. These diverse cell lineages are recapitulated in intestinal organoids, either derived from human induced pluripotent stem cells (hiPSCs) or from intestinal-specific adult stem cells (enteroids) [15,16]. Differentiation of hiPSCs to intestinal organoids (HIOs) results in the generation of intestinal epithelial lineages associated with their mesenchyme [14]. Human enteroids and HIOs have been used to mimic the ex vivo infection of a diverse range of human enteroviral pathogens including rotaviruses, norovirus, and even SARS-CoV-2 [17,18,19,20,21,22]. Enteroids infected with enteroviruses have shown a different tropism for specific epithelial lineages depending on the virus that was tested. For instance, EV-A71 preferentially infects mucin-expressing goblet cells, which reduces the expression of mucin+ cells without affecting the expression of other epithelial lineage markers [23]. Other enteroviruses, such as echoviruses and coxsackieviruses, preferentially infect enteroendocrine cells and enterocytes, while they are unable to replicate in mucin+ goblet cells [20]. Thus, using organoids in the context of viral infection has broadened our understanding of the biology of the host/enterovirus interactions. A similar approach should focus on validating the organoid model to discover and study new antiviral compounds.

Therefore, in this study we used HIOs to study EV-A71 infection and replication in comparison with a standard cell line (RD cell line). To evaluate HIOs as an antiviral platform compared to the RD cell line, we selected three reference compounds (enviroxime, rupintrivir, and 2′-*C*-methylcytidine (2′CMC)) based on their different modes of action, but also based on their available information and potential safety issues when their activity was tested in vivo against enteroviral infections. Enviroxime targets the viral 3A protein, which is essential for the formation of the replication organelles [24]. Rupintrivir inhibits viral 3C protease, which has an important role in virus replication by cleaving viral polyprotein and host protein [25]. A nucleoside analog, 2′CMC targets the viral polymerase and was initially developed as a treatment for hepatitis C virus (HCV) [26]. The toxicity and antiviral activity data of enviroxime, rupintrivir, and 2′CMC presented here show how HIOs are a more sensitive model than RD cell line for determining the potency of antiviral compounds, as well as potential toxic effects.

## 2. Materials and Methods

### 2.1. Human Small Intestinal Organoids and RD Cells

The hiPSC line that was used for this study is NCRM-3 acquired from the Regenerative Medicine Program (RUCDR Infinite Biologics). The primary line was CD34+ from cord blood reprogrammed with a non-integrating episomal plasmid containing POU Class 5 Homeobox 1 (POU5F1), SRY-Box Transcription Factor 2 (SOX2), KLF Transcription Factor 4 (KLF4), and myc proto-oncogene (MYC).

HiPSCs were cultured in colonies on Matrigel (Corning Inc., Corning, NY, USA) in mTeSR1 medium (STEMCELL Technologies, Vancouver, BC, Canada). HIOs were differentiated from the hiPSC line using STEMdiff™ Intestinal Organoid Kit (STEMCELL Technologies) following the manufacturer’s instructions. Briefly, hiPSCs were plated and differentiated towards definitive endoderm (day 3), and then to mid-/hindgut cells (day 5–9 post seeding). The spheroids generated during the mid-/hindgut stage were embedded in Matrigel and grown in intestinal growth media for further differentiation and expansion into intestinal organoids (30–45 days post hiPSCs seeding). After maturation and for long-term culture, HIOs were passaged twice a week, and medium was changed if needed every two days.

RD cells (ATCC, CCL-136, Manassas, VA, USA) were maintained in Dulbecco’s modified Eagle’s medium (DMEM) (Gibco, Thermo Fisher Scientific, Waltham, MA, USA) supplemented with 10% fetal bovine serum (FBS) (Gibco), 1% L-glutamine (Gibco), 1% sodium bicarbonate (Gibco), and 1% penicillin-streptomycin (Gibco). The cell culture was incubated at 37 °C under 5% CO_2_.

### 2.2. Virus Strain and Titration

EV-A71 BrCr strain, a gift from Prof. F. van Kuppeveld (University of Utrecht, Utrecht, The Netherlands), was propagated in RD cells, cultured in 2% FBS-DMEM at 37 °C under 5% CO_2_. The supernatant was harvested when 90% of the cells exhibited cytopathic effect (CPE) and centrifuged to produce virus stock, which was immediately stored at −80 °C until use. EV-A71 stock titer was determined by the 50% tissue culture infective dose (TCID_50_) assay (Reed and Muench, 1938).

### 2.3. Virus Infection of Human Small Intestinal Organoids and RD Cells

To quantify the cell number of HIOs for calculation of multiplicity of infection (MOI), an aliquot of the HIOs fragments was dissociated into single-cell suspensions using 0.05% Trypsin-EDTA (Gibco) for 5 min at 37 °C. The cells were counted using Coulter Counter Z2 (Beckman Coulter Inc., Brea, CA, USA).

For infection, HIOs were mechanically fragmented by vigorously pipetting using a P1000 to expose the apical surface to the virus inoculum as previously described [27,28,29]. After virus inoculation (MOI = 0.01, 0.1, or 1) with EV-A71 or mock infection, the mixture was then incubated for 2 h at 37 °C for virus adsorption. After washing, the infected HIO fragments were mixed with 20% Matrigel-organoid growth medium and seeded at 10,000 cells/well in a 96-well plate pre-coated with 20% Matrigel-organoid growth medium at 50 µL/well.

For RD cell line infection, 20,000 cells/well were cultured in 96-well plate at 37 °C for 24 h. To prepare virus inoculum (MOI = 0.01, 0.1, or 1), EV-A71 was diluted in 2% FBS-DMEM. Medium on cell monolayer was removed, and the inoculum was added for 1 h at 37 °C for virus adsorption. After washing, the monolayer was cultured in DMEM supplemented with 2% FBS.

### 2.4. Viability Assay

Viability of HIOs and RD cells were determined by Cell Titer-Glo^®^ 3D Cell Viability Assay (Promega, Fitchburg, WI, USA) and Cell Titer-Glo^®^ Luminescent Cell Viability Assay (Promega), respectively. According to the manufacturer’s instructions, in a white 96-well plate (Greiner Bio-One, Stonehouse, UK), a 1:1 ratio of reagent and spent culture medium was added to the wells and mixed, and the luminescence signal was measured by CLARIO star plus (B M G Labtech Ltd., Aylesbury, UK).

### 2.5. Compounds, Cytoxicity, and Effective Concentration

EV-A71-infected HIOs and RD cells were treated with rupintrivir (Sigma-Aldrich, St. Louis, MO, USA), enviroxime (aablocks LLC, San Diego, CA, USA), and 2′CMC (Sigma-Aldrich, St. Louis, MO, USA).

For determination of compounds’ half maximal cytotoxic concentration (CC_50_), the non-infected HIOs/cells were treated with compounds (0.5–100 μM) for 72 h. The viability % was calculated as follows: (luminescence reading compound treated)/(luminescence reading DMSO control) × 100. The CC_50_ was calculated using non-linear regression (curve fit) with GraphPad Prism 9 (GraphPad, La Jolla, CA, USA). For determination of compounds half maximal effective concentration (EC_50_), the infected HIOs/cells were treated with DMSO (i.e., 0 µM) or compounds (0.15–40 μM) for 72 h. The CPE inhibition % was calculated as follows: (luminescence reading compound treated)/(luminescence reading DMSO control) × 100. The EC_50_ was calculated using non-linear regression (curve fit) with GraphPad Prism 9 (GraphPad).

### 2.6. RNA Isolation, cDNA, and Quantitative RT-PCR (RT-qPCR)

RNA was extracted from harvested samples using RNeasy Plus Mini Kit (Qiagen, Hilden, Germany). cDNA was created from the isolated RNA using High-Capacity cDNA Reverse Transcription Kit (ThermoFisher Scientific, Waltham, MA, USA) according to the manufacturer’s instructions.

For HIOs cell type expression, each 10 µL reaction mixture contained: 5 µL of TaqMan™ Fast Advanced Master Mix (ThermoFisher Scientific), 3.5 µL of RNase-free water, 1.5 µL of TaqMan gene expression assay (Appendix A; ThermoFisher Scientific), and 1 µL of cDNA as template. The reaction plate was loaded in a LightCycler^®^ 480 Real-Time PCR System (Roche, Basel, Switzerland) and ran at 95 °C for 2 min for pre-incubation, followed by 40 amplification cycles of 95 °C for 1 s and 60 °C for 20 s. Signal detection and measurements were taken in each amplification cycle. RT-qPCR cycle values (C_T_) obtained for specific mRNA expression in each sample were normalized to C_T_ values of human endogenous (housekeeping) gene *EEF1A1* (Eukaryotic Translation Elongation Factor 1 Alpha 1), resulting in the ΔC_T_ values used to calculate relative gene expression (2^−ΔCT^).

For EV-A71 replication quantification, each 10 µL reaction mixture contained: 5 µL of Master Mix, 1.5 µL of RNase-free water, 0.5 µL each of gene-specific forward and reverse primer, and 3 µL of cDNA as template. The reaction plate was loaded in a LightCycler^®^ 480 Real-Time PCR System (Roche) and ran at 95 °C for 3 min for pre-incubation, followed by 40 amplification cycles of 95 °C for 15 s and 60 °C for 60 s. Signal detection and measurements were taken in each amplification cycle. The EV-A71 viral loads in the samples were detected with the PanEV primer using a previously described customized primers probe mix (Integrated DNA Technologies, Coralville, IA, USA) [30]. The viral gene copy number was determined by absolute quantification and making a standard curve using custom gBlocks™ Gene Fragments (Integrated DNA Technologies) (5′-AACACACTACCTTTTGATTCCGCATTGAACCACTGTAATTTCGGACTATTGGTGCCTATCAGCCCGCTGGATTTCGACCAAGGGGCGACACCGGTAATTCCTATCACTATCACGTTGGCTCCGATGTGTTCT-3′).

### 2.7. Immunofluorescence Microscopy

For intestinal cell type expression, HIOs were cultured in 100% Matrigel in black 96-well plates (PerkinElmer Inc., Waltham, MA, USA) and fixed with 4% paraformaldehyde (VWR chemicals) at room temperature for 30 min. For immunofluorescence imaging of infected HIOs, EV-A71 or mock-infected HIOs cultured in slurry of 50% Matrigel in black 384-well plates (PerkinElmer) were fixed with 2% paraformaldehyde (VWR chemicals) at 4 °C overnight. The samples were then washed with PBS and maintained in blocking buffer containing 3% BSA (Sigma-Aldrich, St. Louis, MO, USA), 0.2% Triton X-100 (Sigma), and 1% donkey serum (Merck Millipore, Burlington, MA, USA), for at least 1 h. Subsequently, HIOs were incubated with primary antibodies at 4 °C overnight, washed, and then incubated with Alexa-Fluor-conjugated secondary antibodies (Invitrogen, Waltham, MA, USA) for 1 h at room temperature before washing with PBS. The following antibodies or dyes were used: 4′,6-diamidino-2-phenylindole (DAPI; Invitrogen), F-actin phalloidin conjugated to Alexa Fluor 594 (Invitrogen), lysozyme (Abcam), mucin 2 (BD Biosciences), and anti-dsRNA antibody (clone rJ2, Sigma). Confocal microscopy was performed with IN Cell Analyzer 6000 (GE Healthcare, Chicago, IL, USA) using a 40× objective or Cell Voyager CV 8000 High-Content Screening System using a dry 10× objective. Columbus version 2.9.1 software (Perkin Elmer, Waltham, MA, USA)was used for image analysis.

### 2.8. Statistical Analysis

All data were analyzed with GraphPad Prism version 9.3.1 software (GraphPad, La Jolla, CA, USA). Unpaired *t*-test was used for data analysis. A value of *p* < 0.05 was considered statistically significant; * indicates *p* < 0.05, ** indicates *p* < 0.01, *** indicates *p* < 0.001; and **** indicates *p* < 0.0001.

## 3. Results

### 3.1. Assessment of the Susceptibility of 3D Human Small Intestinal Organoids to EV-A71 Infection

We established a 3D HIO model by differentiating an hiPSC line into small intestinal organoids using a previously described protocol. The presence of intestinal epithelial cell lineages in HIOs was assessed by mRNA expression of specific markers such as Leucine-Rich Repeat-Containing G-Protein-Coupled Receptor 5 (LGR5, stem cells), villin 1 (VIL1, enterocytes), chromogranin A (CHGA, enteroendocrine cells), and lysozyme (LYZ, Paneth cells), or by immunostaining for mucin 2 (MUC2, Goblet cells), lysozyme (LYZ, Paneth cells), villin (VIL1, enterocytes), and chromogranin A (CHGA, enteroendocrine cells). Analysis of the RT-qPCR results shows that HIOs express all the intestinal cell lineages (Figure 1A). Immunostaining of HIOs shows the expression of lysozymes (Figure 1B), enterocytes, and enteroendocrine cells (Appendix A), including goblet cells, which are considered the main cell type infected by EV-A71 (Figure 1B).

Next, HIOs were infected with EV-A71, and the kinetics of viral replication were studied up to 72 h post-infection (hpi). To that end, mechanically disrupted HIOs were infected with EV-A71 at different MOIs (0.01, 0.1, and 1) and monitored daily to check for morphological differences and cytopathic effect (CPE). At 48 hpi, the virus load, determined by RT-qPCR, reached a peak of ~5.5 ± 0.3 log_10_ viral RNA copies/μL at all tested MOIs (Figure 1C). Moreover, distinctive CPE was observed in infected HIOs compared to the mock, starting from 24 hpi to 72 hpi (Figure 1D). This observation was corroborated by a reduced cell viability of the EV-A71-infected HIOs, as determined by an ATP-based colorimetric viability assay, observed from 24 hpi at different MOIs. Figure 1E shows a reduction in HIOs’ viability with increasing MOIs and time post-infection. Finally, the presence of actively replicating virus within EV-A71-infected cultures was confirmed by immunostaining against the dsRNA intermediate, which is only present during the genome replication of (+) ssRNA viruses such as EV-A71 (Figure 1F). Thus, we confirmed that the HIO line supports the replication of EV-A71, leading to a complete CPE after 72 hpi, in an MOI-dependent manner.

### 3.2. Comparison of EV-A71 Replication Kinetics between HIOs and RD Cells

To compare the replication of EV-A71 in 3D HIOs versus a two-dimensional (2D) model, we used RD cell line, which is commonly used to assess EV-A71 replication in vitro. Virus replication in RD cells was analyzed at 24, 48, and 72 hpi using the same MOIs previously tested in the HIOs experiments, and RT-qPCR was used as a readout to quantify the virus yield (Appendix A). Within 24 hpi, there was an increase in viral RNA copies in RD cells, but this was lower than that detected in HIOs at all MOIs tested (Figure 2A). Likewise, cell viability decreased less at early time-points post-infection, when compared to HIOs (Figure 2B). A delay in the replication kinetics of EV-A71 replication in RD cells could be the underlying reason for this. A higher peak of replication was reached at 48 hpi, when cell viability also started to significantly decrease (Appendix A). Taken together, infection of RD cells with EV-A71 showed a direct correlation of cell viability and viral load detected by RT-qPCR, like in the HIOs model.

### 3.3. Compounds Cytotoxicity in HIOs and RD Cells

Having established and compared the conditions to infect either HIOs or RD cells with EV-A71, we next tested the antiviral activity of three different antivirals with distinct mechanisms of action (enviroxime, rupintrivir, and 2′CMC) against EV-A71 in HIOs and RD cell line models. The results are presented in Figure 3 and Table 1. The cytotoxicity of enviroxime, rupintrivir, and 2′CMC was the first parameter assessed. HIOs or RD cells were exposed to a range of concentrations of each compound for 72 h. Treatment with enviroxime at 100 µM reduced the cell viability of HIOs to 0%, resulting in a calculated CC_50_ of 24 ± 7 µM. In RD cells, the viability was reduced to 20% at the highest tested concentration (100 μM), and the calculated CC_50_ is 28 ± 2 µM. Treatment with rupintrivir did not result in cytotoxicity in HIOs, nor in RD cells, at all tested concentrations, and therefore CC_50_ is higher than 100 µM. Finally, HIOs treated with 2′CMC showed a reduction of viability at the maximum tested concentration of 100 μM with a determined CC_50_ of 48 ± 1 µM. On the other hand, 2′CMC did not affect the viability of RD cells at the tested concentrations (CC_50_ > 100 µM). Altogether, these results suggest that there are differences in the cytotoxicity of the tested compounds in HIOs when compared to a commonly used immortalized cell line.

### 3.4. Comparison of Antiviral Activity Determined by CPE Inhibition and Viral RNA Yield between HIOs and RD Cells

Next, we studied the antiviral effect of the molecules in both HIOs and RD cells upon infection with EV-A71 at MOI 0.1. Infected HIOs/cells were exposed to enviroxime, rupintrivir, or 2′CMC for 72 h. While we observed EV-A71-induced CPE in HIOs and RD cells upon visual inspection, this was reduced by treatment with 1.25 μM of enviroxime or rupintrivir in both HIOs and RD cells (Appendix A). To investigate whether there was still virus replication upon antiviral activity, HIOs and RD cells were immunofluorescent labeled with dsRNA (Appendix A).

Enviroxime showed antiviral activity in HIOs, with an EC_50_ of 0.4 ± 0.2 μM and 1.4 ± 0.3 μM, determined by CPE inhibition and RNA yield reduction, respectively. By contrast, the determined EC_50_ in the RD cells was 10-fold lower (EC_50_ = 0.06 ± 0.001 µM and 0.2 ± 0.04 μM, respectively) (Table 1).

Rupintrivir showed a strong antiviral activity in both HIOs and RD cells, resulting in full protection from EV-A71-induced CPE at the lowest concentration tested of 0.15 μM and 0.035 μM, respectively. Still, by looking at virus yield data (Figure 4B, middle panel), there was a concentration-dependent reduction in virus RNA level in HIOs (EC_50_ = 1.7 ± 0.4 μM), while in RD cells, RNA levels were lowered to 10 ± 3% at all tested concentrations (EC_50_ < 0.035 μM). We additionally looked at whether the less pronounced viral RNA yield reduction observed in HIOs was due to the incomplete inhibition of viral replication by rupintrivir. Visual inspection showed that the viability of rupintrivir-treated HIOs was similar to the mock, indicating that the drug treatment rescued HIOs from EV-A71-induced CPE (Figure 4C). While some virus (dsRNA marker) was still detectable in the epithelial cells, the signal was much reduced and confined to single cells (Figure 4D).

Finally, after treatment with 2′CMC, there was a reduction of virus-induced CPE of EV-A71-infected HIOs of up to 66 ± 17 %, as determined by an ATP-based viability assay (Figure 4A, right panel). A plateau of CPE reduction was observed from 2.5 μM up to the highest tested concentration (20 μM). Treatment of EV-A71-infected RD cells resulted in complete protection from virus-induced CPE (EC_50_ = 1.3 ± 0.04 μM). Furthermore, 2′CMC treatment showed a concentration-response reduction of the detected viral RNA in HIOs and RD cells (Figure 4B, right panel). While the calculated EC_50_ values of 2′CMC in HIOs and RD cells were comparable (Table 1), the antiviral showed poorer selectivity in the HIO model.

## 4. Discussion

In this study, we differentiated hiPSCs to HIOs and used them as a model to study EV-A71 infection and antiviral treatment. HIOs contain all cell types typically found in human small intestine, including stem cells, enterocytes, Paneth cells, and goblet cells. We showed that the 3D-HIOs model is amenable to infection and allows for the replication of EV-A71. To evaluate the HIO model as an ex vivo platform to study infection with enteroviruses and its potential use to test the effect of antiviral treatments, we compared the performances of three different antiviral drugs between HIOs and RD cell line infected with EV-A71.

First, we compared the kinetics of EV-A71 replication in HIOs and RD cells. In HIOs, there was a higher virus yield at 24 hpi than in the cell line (Figure 2A). A previous study showed a higher mRNA transcript expression level of the EV-A71 scavenger receptor class B member 2 (SCARB2) and P-selectin glycoprotein ligand-1 (PSGL-1) in enteroids compared to RD cell line [21], which might contribute to a faster internalization and replication of EV-A71 in HIOs. The higher virus yield reached in HIOs also translated into a more rapid decline of viability of the infected HIOs compared to RD cells, especially at higher MOIs.

Next, we tested three compounds with antiviral activity against multiple enteroviruses and different viral targets: enviroxime, rupintrivir, and 2′CMC. Enviroxime and enviroxime-like compounds were previously tested in vitro against EV-A71 and have shown inhibitory effects [31,32,33,34]. In this study, we tested the inhibitory effect of enviroxime against EV-A71 in an organoid model for the first time. Although the compound was still active in HIOs, a 10-fold higher EC_50_ than RD cells was found in this more complex model system, also resulting in a poor selectivity index (Table 1). This suggests the cell line system has a lower “barrier” for small molecules to show activity. In line with these results, previous clinical studies have reported gastrointestinal side effects and no significant clinical effects of enviroxime against infection with human rhinovirus (HRV) [35,36].

Rupintrivir was developed by Pfizer, and the phase II clinical studies against HRV showed no adverse side effects or treatment effects [37]. Unfortunately, further development of rupintrivir as an anti-HRV treatment was halted due to lack of efficacy in natural infection cases. Nevertheless, rupintrivir was explored as an antiviral against other enteroviruses such as EV-A71 and showed strong virus inhibition [38,39,40]. In the current study, there was no cytotoxicity by rupintrivir at the tested concentrations in HIOs nor in RD cells (CC_50_ > 100 µM). While there was complete CPE inhibition (based on viability) and normal morphology was observed at all tested concentrations in both models, there was a concentration-dependent decrease of viral RNA yield in EV-A71-infected HIOs. This suggests that a higher drug exposure of rupintrivir was required to inhibit all viral replication in HIOs, when compared to the RD cells. Interestingly, previous in vivo studies of rupintrivir as an antiviral agent showed a complete inhibition of symptoms promoted by the infection upon rupintrivir treatment, while the virus load was decreasing in a similar stepwise manner as we observed in HIOs [41,42].

While the antiviral effect of 2′CMC has been tested against many other enteric viruses, we studied its activity against EV-A71 here [43]. Valopicitabine, the oral valine ester prodrug of 2′CMC, exhibited gastrointestinal side effects during a clinical study against HCV infection, after 8 to 12 weeks of treatment [44,45,46]. In this study, we detected some toxicity due to 2′CMC (CC_50_ = 48 ± 1 µM) in the HIOs model, but not in RD cells (CC_50_ > 100 µM). Furthermore, 2′CMC-treated infected HIOs did not display 100% viability, even at the highest 2′CMC concentration tested. This could imply some low-grade harmful effect on the host cell. Altogether, the calculated SI for 2′CMC in HIOs was rather poor compared to the RD cell line model (Table 1).

Overall, two of the three compounds selected in this study show a lower SI in the HIO model, which seems in line with pre-clinical in vivo data for these compounds. These results imply that the HIO model could play an important role in finding a more refined selection of pre-clinical candidates to be tested in animal models and in the clinic. In fact, enhanced cytotoxicity was also reported for other drugs such as itraconazole when tested in enteroids compared to cell lines [21]. This, combined with the lower sensitivity to the antiviral drug treatment of infected HIOs, suggests that using this model might in the future improve pre-clinical candidate selection and reduce the current high attrition rate in the development of new potential antiviral drugs [47].

Historically, CPE-based readouts have been used to assess antiviral compound activity, constituting a high-barrier and multiple replication cycle readout that allowed the identification of multiple classes of inhibitors. Yet, it is a one parameter/one timepoint readout that does not provide detailed information on the effect of the compound on the virus and host cell. This is highlighted by the results of rupintrivir treatment in infected HIOs, where the CPE protective effect was present, but a concentration-dependent effect on viral RNA yield was detected and confirmed by staining for replicating virus. This exemplifies how a more complex biological model offers the possibility to unveil additional layers of the host/pathogen interactions. In this respect, combination of organoid models with the analysis of multiparameter readouts will provide additional levels of information regarding antiviral activity and cytotoxicity to support a better selection and prioritization of “hit” antiviral compounds.

Another advantage of organoid models is the ability to sustain the cultivation of clinical isolates, thus facilitating the validation of lead compounds in patient-derived samples rather than in cell-culture-adapted virus strains [22,48]. Thus, further anti-EV-A71 studies in HIOs using clinically isolated virus might provide further insight on multiple aspects of virus infection, in addition to strengthening efforts to develop effective treatments. It would also be interesting to explore, in HIOs, compounds that specifically target host cells in treating viral infections, especially those that have been previously tested in cell lines and failed [49].

## 5. Conclusions

In summary, we studied EV-A71 infection and replication in HIOs in the context of antiviral drug discovery. Our results demonstrate the added value of using such an HIO model in antiviral studies over the traditional immortalized cell lines. We also showed that 2′-*C*-methyl nucleosides could be developed as antivirals against EV-A71 infections.

## Figures and Tables

**Figure 1 cells-12-01138-f001:**
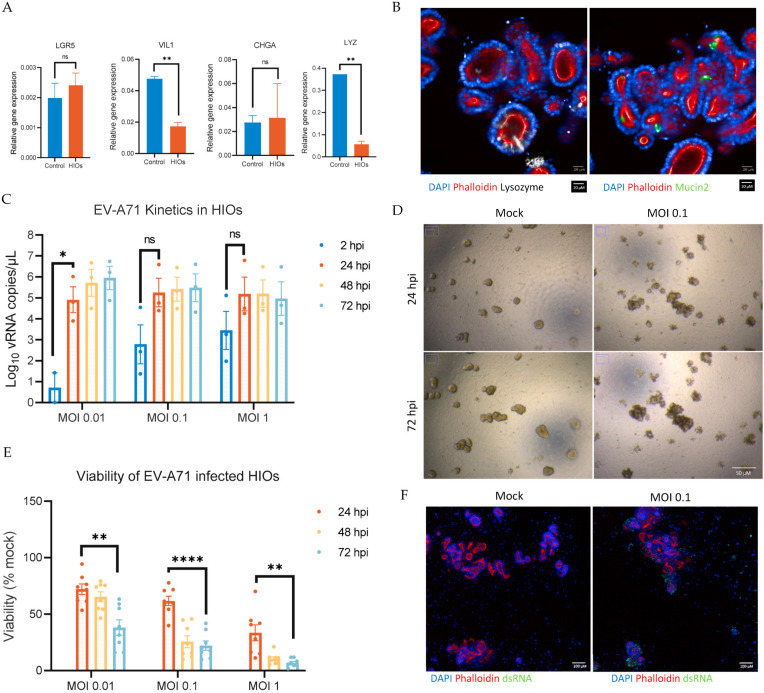
Characterization of the differentiation of human induced pluripotent stem-cells-derived small intestine organoids (HIOs) and Enterovirus A71 (EV-A71) replication kinetics in the derived HIOs and viability post infection. (**A**) Relative expression of total RNA of adult small intestine as a control and HIOs. The expression level of each gene was calculated relative to EEF1A1 gene expression as housekeeping gene. (**B**) Images represent differentiated HIOs: DAPI (blue) for nucleus, phalloidin (red) for F-actin, lysozyme (white) for Paneth cells, and mucin 2 (green) for goblet cells. Scale bar, 20 µM. (**C**) HIOs were infected with EV-A71 virus at multiplicity of infection (MOI) 0.01, 0.1, and 1. At the indicated time points, total culture was collected by combining the triplicates to measure the levels of viral RNA yield by RT-qPCR. (**D**) Brightfield images of mock HIOs and MOI 0.1 infected HIOs at 24 h post-infection (hpi) and 72 hpi show a difference in morphology indicating cytopathic effect (CPE) due to infection. All images were acquired using the same objective. Scale bar, 50 µM. (**E**) To measure viability of HIOs, Cell Titer-Glo 3D was used at the indicated time points post-infection. Mock-infected HIOs were used to normalize the viability as a percentage. (**F**) Images represent mock- and EV-A71-infected HIOs (0.1 MOI) at 24 hpi: DAPI (blue) for nucleus, phalloidin (red) for F-actin, and dsRNA (green). Scale bar, 100 µm. Data are the mean ± SEM of three independent experiments, each carried out in triplicate; * indicates *p* < 0.05, ** indicates *p* < 0.01, and **** indicates *p* < 0.0001; ns indicates not significant.

**Figure 2 cells-12-01138-f002:**
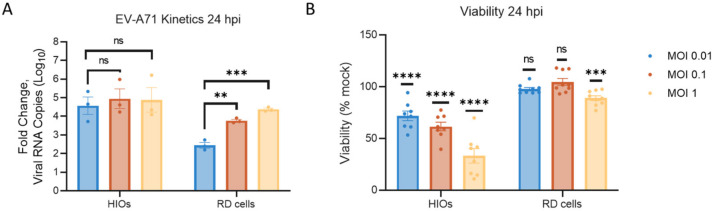
Comparison of EV-A71 replication kinetics between HIOs and RD cells. (**A**) EV-A71 replication kinetics 24 hpi. HIOs and RD cells were infected with EV-A71 virus at 0.01, 0.1, and 1 MOI. Total culture of HIOs and supernatant of RD cells were collected by combining triplicates and measured for viral RNA yield by RT-qPCR, and fold change is calculated based on mock infection. (**B**) Viability 24 h post EV-A71 infection. Viability in HIOs and RD cells was measured using Cell Titer-Glo 3D or Cell Titer-Glo, respectively. Mock-infected HIOs/cells were used to normalize the viability as a percentage. Data are the mean ± SEM of three independent experiments, each carried out in triplicate; ** indicates *p* < 0.01, *** indicates *p* < 0.001, and **** indicates *p* < 0.0001; ns indicates not significant.

**Figure 3 cells-12-01138-f003:**
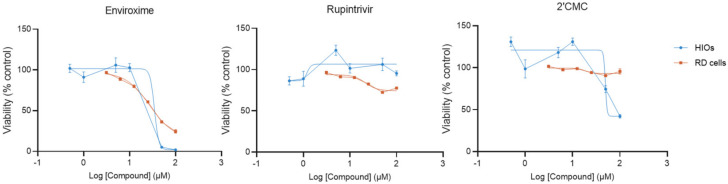
Compounds’ cytoxicity in HIOs and RD cells. To measure the viability of HIOs and RD cells, Cell Titer-Glo 3D and Cell Titer-Glo, respectively, were used 72 h post-treatment. DMSO control was used to normalize the viability as a percentage. The half maximal cytotoxic concentration (CC_50_) was calculated using Prism nonlinear regression (GraphPad Prism 9). Data are the mean ± SEM of at least four independent experiments, each carried out in triplicate.

**Figure 4 cells-12-01138-f004:**
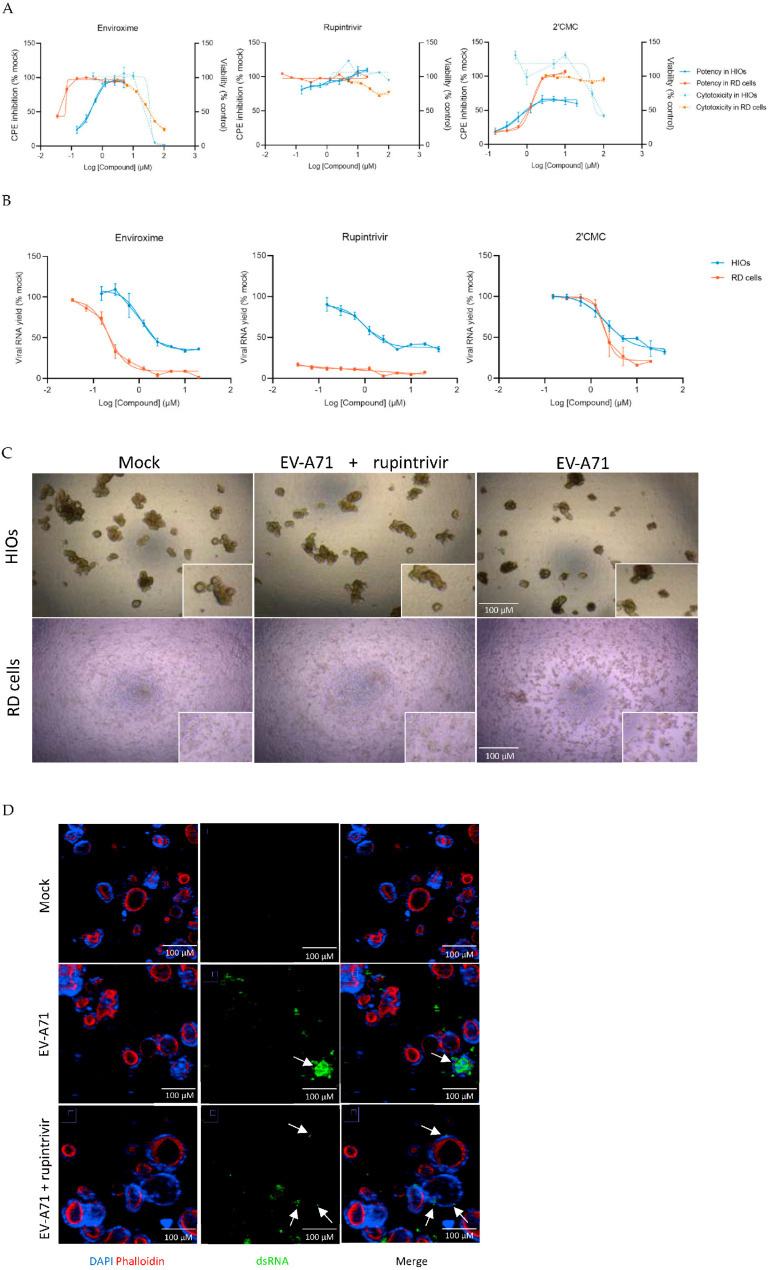
Comparison of antiviral activity determined by CPE inhibition and viral RNA yield between HIOs and RD cells. (**A**) Viability of HIOs and RD cells was measured, using Cell Titer-Glo 3D or Cell Titer-Glo, after 72 h of EV-A71 infection at MOI 0.1 and incubation with compounds at indicated concentrations. CPE inhibition is calculated based on percentage of mock-infected control. The half maximal effective concentration (EC_50_) was calculated using Prism nonlinear regression (GraphPad Prism 9). (**B**) For viral RNA yield, HIOs and RD cells were infected with EV-A71 at an MOI of 0.1 and treated with compounds at indicated concentrations for 72 h before total culture of HIOs or supernatant of cells were collected by combining triplicates and determined by RT-qPCR. Viral RNA yield is calculated based on percentage of mock-infected control. The EC_50_ was calculated using Prism nonlinear regression (GraphPad Prism 9). (**C**) Brightfield images of mock control, infection treated with 1.25 µM rupintrivir, and infected control HIOs and RD cells at 72 hpi. All images were acquired using the same objective. Scale bar, 100 µM. (**D**) Images represent mock-, EV-A71-infected HIOs (0.1 MOI), and EV-A71-infected HIOs (0.1 MOI) treated with rupintrivir (10 µM) at 48 hpi: DAPI (blue) for nucleus, phalloidin (red) for F-actin, and dsRNA (green). Scale bar, 100 µM. Data are the mean ± SEM of at least six independent experiments; each carried out in triplicate.

**Table 1 cells-12-01138-t001:** CC_50_, EC_50_, and selectivity indices of enviroxime, rupintrivir, and 2′CMC in HIOs and RD cells.

Compounds	Enviroxime			Rupintrivir			2′CMC		
	CC_50_ (μM)	EC_50_ (μM)	SI *	CC_50_ (μM)	EC_50_ (μM)	SI	CC_50_ (μM)	EC_50_ (μM)	SI
HIOs	24 ± 7	0.4 ± 0.2	60	>100	<0.15	>667	48 ± 1	1.0 ± 0.3	48
		1.4 ± 0.3	17		1.7 ± 0.4	>59		1.5 ± 0.3	32
RD cells	28 ± 2	0.06 ± 0.001	467	>100	<0.035	>2857	>100	1.3 ± 0.04	>76
		0.2 ± 0.04	140		<0.035	>2857		2 ± 0.4	>50
Significance	ns	*p* = 0.037		NA	NA		NA	ns	
		* p * = 0.010			NA			ns	

* Selectivity Index (SI) = CC_50_/EC_50_. EC_50_ values in black were based on CPE inhibition data; EC_50_ values in blue were based on RNA yield data. Values are expressed as mean ± SE. ns indicates not significant; NA indicates not applicable.

## Data Availability

The data presented in this study are all presented in the manuscript.

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
