# Peer review of "Evaluation of 3D Human Intestinal Organoids as a Platform for EV-A71 Antiviral Drug Discovery"

_cells, 2023, doi:10.3390/cells12081138_

Round 1
Reviewer 1 Report
This manuscript demonstrates comparative studies of EV-71 infection in the RD cell model and HIO model. The result indicates that the HIO model is more sensitive to infection and drug treatment than the RD cell model. The manuscript is well-written, and the data is clearly presented. The manuscript can be further improved by addressing the below issues,
(1) Epithelial cells in HIO are polarized. The apical side is inside the organoid lumen, while the basal side is outside. However, the critical receptors for infection are usually located on the apical side. Thus, the HIO is supposed to be more resistant to infection. The authors should discuss their finding that the HIO model is more sensitive to infection, as it is controversial.
(2)it will be interesting also to investigate intestinal inflammation post-infection.
(3) the authors should provide high-resolution fluorescence images for Mucin2, Lysozyme, and DsRNA stainings.
(4) RD cells were not well grown, as illustrated in Figure 4C. Thus, it's not fair to compare these data to that in the HIO group.
(5) It is also vital to provide fluorescence staining data for the RD cell group in Figures 1B&F, and 4D.
(6) Scalar bars need to be added to all the photos.
Reviewer 2 Report
The authors have modeled EV-A71 infection of 3D human intestinal organoids and screened several inhibitors. This provides a new approach to the field of drug development and personalized medicine. In general, this paper has a relatively rich set of experimental data. All the data could well support the point of view of this manuscript. However, some issues with the study need to be addressed before it can be considered for publication.
1. A certificate of appropriate ethical review should be provided for the hIPSC used in this study.
2. The experimental methods were not cited the references. For example, in infection experiments, intestinal organoids had to be mechanically sheared before virus inoculation.
3. If possible, immunofluorescence double staining should be added for all cell markers and viruses, including: enterocyte marker villin (VIL1), intestinal alkaline phosphatase (ALPI), Paneth cell marker lysozyme (LYZ), goblet cell marker mucin 2 (MUC2) and enteroendocrine cell chromogranin A (CHGA). The in vitro culture time and the number of generated intestinal organs should be described.
4. Fig. 3, the picture is obscured and needs to be recalibrated.
5. Fig. 4A-B, P values of all comparisons should clearly indicated in the Figures.
6. Fig. 4 should display bright-field images and immunofluorescence images of all inhibitors before and after treatment to show the significant advantages of rupintrivir.
7. Fig. 4 D-E, the corresponding scales need to be added in all the picture.
